# Novel Anterior Cranial Base Area for Voxel-Based Superimposition of Craniofacial CBCTs

**DOI:** 10.3390/jcm11123536

**Published:** 2022-06-20

**Authors:** Georgios Kanavakis, Mohammed Ghamri, Nikolaos Gkantidis

**Affiliations:** 1Department of Pediatric Oral Health and Orthodontics, UZB-University Center for Dental Medicine, University of Basel, 4058 Basel, Switzerland; georgios.kanavakis@unibas.ch; 2Department of Orthodontics, Tufts University School of Dental Medicine, Boston, MA 02111, USA; 3Department of Orthodontics and Dentofacial Orthopedics, University of Bern, 3010 Bern, Switzerland; mghamri@moh.gov.sa; 4Directorate of Health Affairs-Jeddah, Ministry of Health, Riyadh 11176, Saudi Arabia

**Keywords:** imaging, three-dimensional, cone-beam computed tomography, voxel-based superimposition, anterior cranial base

## Abstract

A standard method to assess changes in craniofacial morphology over time is through the superimposition of serial patient images. This study evaluated the reliability of a novel anterior cranial base reference area, principally including stable midline structures (EMACB) after an early age, and compared it to the total anterior cranial base (TACB) and an area including only midline structures (MACB). Fifteen pairs of pre-existing serial CBCT images acquired from growing patients were superimposed with all techniques by applying a best-fit registration algorithm of corresponding voxel intensities (Dolphin 3D software). The research outcomes were the reproducibility of each technique and the agreement between them in skeletal change detection, as well as their validity. The TACB and EMACB methods were valid, since the superimposed midline ACB structures consistently showed adequate overlap. They also presented perfect overall reproducibility (median error < 0.01 mm) and agreement (median difference < 0.01 mm). MACB showed reduced validity, higher errors, and a moderate agreement to the TACB. Thus, the EMACB method performed efficiently and mainly included the stable midline ACB structures during growth. Based on the technical, anatomical, and biological principles applied when superimposing serial 3D data to assess craniofacial changes, we recommend the EMACB method as the method of choice to fulfil this purpose.

## 1. Introduction

Morphological changes in the craniofacial structures occur during growth and development and as a result of orthopedic and craniofacial treatments. These concepts apply in orthodontics, orthognathic surgery, and other related specialties. In order to describe and quantify such changes, the most common method applied is the superimposition of serial cephalometric images on the anterior cranial base [1,2]. The development of the anterior cranial base is completed at an early age and its structures remain stable thereafter. Moreover, it has a central location in the craniofacial complex and it is clearly depicted on lateral cephalometric images. Thus, it is considered the golden standard for the superimposition of serial images [3,4]. Despite its broad use, cephalometric imaging has several limitations related to landmark identification, operator error, the magnification of structures, and image distortion related to the depiction of 3D structures on a 2D image [1,2,5,6]. Therefore, its reliability in evaluating morphological changes is questionable.

With the introduction of Cone Beam Computer Tomography (CBCT) as diagnostic tool for the craniofacial region, the limitations of cephalometry have been notably reduced. Using CBCT images eliminates the magnification error and the error due to the compression of 3D structures into a 2D image. CBCTs deliver adequate image quality, have low radiation compared to conventional CT scans, and provide three-dimensional information of the craniofacial structures, thus constituting an effective diagnostic tool for clinical assessments [1,7,8]. In recent years, CBCT images have also been used to perform superimpositions of the craniofacial structures in three-dimensions. For this purpose, the same anatomical guidelines are used with the anterior cranial base also being the structure of refs. [7,9,10].

Three-dimensional superimpositions of CBCTs can be conducted with landmark-based, surface-based, and voxel-based methods. Landmark-based methods require a considerable number of landmarks to be accurate and are highly operator-dependent, therefore they are not cost-effective for clinical use [11,12]. Surface-based methods require a preliminary bone segmentation process to extract surface data from the radiographic volume. Thus, they are highly dependent on the threshold values used for this segmentation process [13,14]. In addition, accurate bone segmentation is difficult to achieve, even if the threshold value remains stable, because the greyscale values in CBCTs do not directly correspond to Hounsfield units as in CT images and therefore the grey level intensities may not be consistent [15]. For voxel-based superimpositions, the reference structures are selected directly on the volumetric image and superimposition is performed through a best-fit approximation of corresponding voxel greyscale values. This method does not include the shortcomings of the other two; however, in order to receive an interpretable three-dimensional result, surface models need to be extracted from the volumetric data, a process which also requires volume segmentation. Nevertheless, the segmentation is only done for visualization purposes and does not affect the superimposition process.

In order to perform accurate assessments when superimposing three-dimensional structures, the superimposition outcomes should be interpreted according to the selected area that serves as the reference for the registration of serial data. The distance between the reference area and the measurement area [11,16,17], the anatomical accuracy of the three-dimensional renderings of the reference area, and the area of interest [14,18], as well as the morphological stability of the reference area over time [9,18,19], should be considered when selecting the reference area. For voxel-based superimpositions of serial CBCTs, the most widely researched and used area includes the central anterior cranial base structures and extends laterally to those structures, as described in previous investigations [20,21,22,23,24]. This area has been shown to be highly reproducible [20,21,22,23,24] and can be used for treatment outcome assessments or the evaluation of craniofacial changes in growing individuals [21,22,23,24]. However, this area includes soft and hard tissues that extend beyond the stable midline anterior cranial base (ACB) structures after an early age [3,4]. This has been addressed by a previous study proposing a smaller area including only the midline ACB structures to be used for the voxel-based superimposition of serial CBCTs [25]. However, this area has shown reduced trueness and reproducibility, which may be attributed to its very small size, allowing higher range of superimposition error [23].

In light of these recent findings, a new area is proposed here for the voxel-based superimpositions of CBCT volumes. This novel area contains the robust midline ACB structures and extends slightly laterally to include the width of the anterior clinoid process of the sphenoid bone. This small extension may potentially counterbalance the inaccuracies stemming from the very narrow axial dimension of the previously proposed central reference area (MACB) [23,25], while still principally including the stable midline anterior cranial base structures. The present investigation evaluated the reliability and reproducibility of a novel Extended Middle Anterior Crania Base area (EMACB) and compared it to the commonly used Total Anterior Cranial Base area (TACB), as well as to the smaller MACB area, including only midline anterior cranial base structures. We hypothesized that the novel EMACB area is valid, shows adequate agreement to the TACB method, and adequate reproducibility when applied on serial CBCT images of growing patients.

## 2. Materials and Methods

### 2.1. Study Design

This research illustrates a prospective methodological study, employing pre-existing patient data.

### 2.2. Sample

The research sample consisted of serial craniofacial CBCT images of 15 orthodontic patients (8 males, 7 females). All images were obtained from patients where 3D information facilitated accurate clinical diagnosis, such as in cases with impacted teeth. In all cases there were adequate clinical indications to justify the use of CBCT imaging. The sample size was considered adequate based on previous similar investigations [21,22,23]. The mean age of the participants at T0 (time of 1st CBCT) was 11.75 ± 0.59 (range: 11.0–12.8) years with a time lap between T0 and T1 (time of 2nd CBCT) of 1.69 ± 0.37 years. Subjects with congenital malformation, systemic diseases, or syndromes that could affect the facial morphology, as well as individuals with extreme facial asymmetries, were excluded. Low-quality scans and images with metallic materials that caused considerable artifacts were also excluded. Two researchers (M.G and N.G.) visually inspected all criteria independently to assess eligibility.

### 2.3. Generation of CBCTs

All tested CBCTs were acquired in a single orthodontic clinic between 2008 and 2018, using the same CBCT machine (KaVo 3D eXam, Hatfield, PA 19440, USA) under the following settings: 170 mm height × 232 mm diameter field of view, 0.4 mm^3^ voxel size, 5 mA tube current, 120 kV tube voltage, 8.9 s scan time, 3.7 s exposure time. These settings allowed for lower radiation doses [26]. The CBCT volumes were exported in a DICOM format for further analyses.

### 2.4. Superimposition Process and Reliability Assessment

Voxel-based superimposition of serial CBCTs was performed with Dolphin 3D software© (Version 2.1.6079.17633, Dolphin Imaging and Management Solutions, Chatsworth, CA 91311, USA).

The pairs of DICOM datasets, acquired at two different time points (T0 and T1), were imported into the software. For the purpose of this study, three superimposition reference areas were compared. The first area, considered as the reference method, was the total anterior cranial base (TACB), the second area was an extended version of the middle anterior cranial base area (EMACB), and the third area was limited to the middle anterior cranial base area (MACB), as described previously [23,25]. All areas include the midline structures of the anterior cranial base that are the standard structures used to assess craniofacial changes [3]. In order to define these areas, the selection frame tool of Dolphin software was used, provided under the option for the manual superimposition of volumes. The anterior–posterior and superior–inferior borders of the TACB and the EMACB areas were identical and delineated by the posterior wall of sinus frontalis (anteriorly), the middle of sella turcica (posteriorly), and a line 2–4 mm inferiorly to the floor of sella turcica. The height of the frame was 3–4 cm, and this defined the superior border of the reference area. The two reference areas differed only in their lateral extensions. The TACB extended laterally to include the entire width of the anterior cranial wall, as described in previous work [20,21,22,23], while the EMACB extended laterally to include the width of the anterior clinoid processes of the sphenoid bone. The MACB area was defined similarly anteroposteriorly, extended less laterally to include only midline structures, and was approximately 1 cm shorter vertically. This area has been thoroughly described and tested previously [23,25], and thus it will be presented here briefly, to allow direct comparison to the two primary study methods (TACB and EMACB). The TACB and EMACB frames are presented in Figure 1.

The reference structures were selected on the base volume (CBCT T0) in a multiplanar view. Adjustment and approximation of CBCT T1 was performed manually to the base volume (CBCT T0) and was then automatically superimposed on it by applying the software’s best fit approximation algorithm, which aims to achieve the best match of greyscale intensity values of the selected voxels. A two- or three-time repetition of the automated voxel-based registration was performed until there was no visible change in the orientation of the superimposed images. For the EMACB and TACB methods, the overlap of the anterior cranial base reference structures was evaluated visually in three planes of space (axial, sagittal, coronal), on the 2D planar DICOM images, to assess the superimposition outcome. The outcome was considered satisfactory when perfect overlap was evident at the superimposed midline anterior cranial base structures. The reoriented position of the superimposed CBCT T1 was saved, and the final visual assessment of the overlap of the stable anterior cranial base structures was recorded as a reliability measure. The reliability of MACB method has been published previously [23].

### 2.5. Measurement Process

Superimposition was done using each reference area (TACB, EMACB, and MACB) independently, and then an automated bone segmentation function of the Dolphin software was used to extract hard tissue surfaces from the T0 and T1 volumes. For further assessment, the extracted models were saved as STL files and imported in Viewbox 4 Software (version 4.1.0.1, BETA 64; dHAL software, Kifisia, Greece). In order to quantify differences in T0–T1 changes between and within methods, and to determine superimposition error, the following measurement areas were used: N-point, A-point, Pogonion, Zygomatic arch right and left, and Gonial angle right and left. The size of each area was 100 triangles. To eliminate the effect of the measurement area selection factor on outcomes, the seven measurement areas were selected once for every subject at the T0 surface model. The subsequent 3D model, including the selected measurement areas, was duplicated and used for all outcomes measured in the study. The exact processes of volume superimposition, surface model extraction, and measurement have been previously described in detail [21,22,23].

There is only one difference in the measurement process used here. To calculate the T0–T1 distances, the T1 anatomical surfaces that corresponded to the seven T0 measurement areas were extracted from the original T1 model through a selection of only the outer T1 skeletal surfaces (Appendix A). This was performed to avoid miscalculation of the distances between corresponding closest points of the T0 and T1 models, because of a lack of anatomical correspondence. This could result from intermediate surfaces created between the outer skeletal surfaces during bone segmentation (Appendix A) or from the large rotational differences of the T0 and T1 models after superimposition. Both conditions could bias the automatic corresponding closest point identification by the software, resulting in measuring distances between noncorresponding anatomical points.

### 2.6. Intraoperator Reproducibility of Superimposition Methods

In order to test reproducibility of the TACB, EMACB, and MACB methods, the entire T0–T1 superimposition process was performed twice by one trained operator (M.G.). STL files of all hard tissue surface models were imported into Viewbox 4 software following each superimposition, and the mean absolute distances (MAD) between corresponding T0–T1 surface models at the seven measurement areas were measured and compared. To provide a visual presentation of the EMACB results, color-coded maps of selected cases were used showing the distances between the repeatedly obtained T1 surface models, with the T0 surface held constant in space.

### 2.7. Agreement between TACB, EMACB, and MACB

In order to assess agreement between the EMACB and MACB methods with the TACB method, and thereby evaluate the reliability of the novel EMACB reference area, the T0–T1 distances between corresponding superimposed 3D models, at the seven pre-determined measurement areas, were compared. The agreement between EMACB and TACB methods was also assessed visually through color-coded maps, as described above, for reproducibility assessment. Zero distance on the color-coded map would indicate perfect agreement between methods.

### 2.8. Statistical Analysis

Statistical analysis was carried out with SPSS Software (IBM SPSS Statistics for Windows, Version 28.0. IBM Corp., Armonk, NY, USA), which was also used to create all relevant graphic representations of the results.

Data normality was tested with the Shapiro–Wilk test and was not consistently present, indicating the use of nonparametric statistical tests. Intraoperator reproducibility on EMACB, TACB, and MACB superimposition outcomes was shown with box plots, where any deviation from 0 indicates superimposition error. Differences in the overall amount of error between EMACB, TACB, and MACB were tested through Kruskal–Wallis test, followed by Dunn’s test for pairwise comparisons, with the *p*-values adjusted by the Bonferroni correction for multiple tests. Following EMACB and TACB superimposition, differences in the amount of error (intraoperator reproducibility) among the different measured areas were tested in a paired manner through Friedman’s test. In case of significant results, pairwise comparisons were performed through Dunn’s test, as above.

Differences in the detected T0–T1 changes between the EMACB and the TACB superimposition were visualized and tested in a similar manner.

In all cases, a two-sided significance test was carried out at an alpha level of 0.05. In case of multiple comparisons, where applicable, a Bonferroni correction was applied to the level of significance to avoid false positive results.

The Bland–Altman method (difference plot) [27] was also used to evaluate intraoperator reproducibility in the detected T0–T1 morphological changes through EMACB, as well as the agreement between EMACB and TACB superimpositions. Assessments of the intra- and interoperator reproducibility of the TACB and MACB methods using Bland–Altman plots have been published previously [21,22,23]. A one sample *t*-test was used to detect the presence of systematic error between the compared measurements.

## 3. Results

### 3.1. Reliability of Superimposition Methods

Visual assessment of the 2D DICOM images of all 15 cases, at all three planes of space, as observed on the screen, showed adequate overlap of the midline anterior cranial base superimposition reference structures that are considered stable during growth, for both TACB and EMACB techniques (Figure 1). The findings were confirmed during the repeated assessments by the same operator. Thus, TACB and EMACB methods showed perfect reliability, in contrast to reduced reliability of MACB that has been published previously [23].

### 3.2. Intraoperator Reproducibility of Superimposition Methods

There were no systematic differences in the T0–T1 changes detected following repeated applications of the TACB, the MACB, and the EMACB method by same operator (one sample *t*-test: *p* > 0.05). The median error of the EMACB superimposition was 0.03 mm (IQR: 0.22 mm), that of the TACB method was −0.01 mm (IQR: 0.15 mm), and that of MACB method was −0.05 mm (IQR: 0.37 mm) (Kruskal–Wallis test: *p* = 0.016). Pairwise comparisons revealed significant differences between the MACB and EMACB method (Dunn’s test: *p* = 0.019, including Bonferroni adjustment). Both EMACB and TACB errors were considered negligible and of no clinical significance, whereas the MACB error could be considered as clinically significant.

The reproducibility of all methods when repeated by the same operator is presented in Figure 2. When using the EMACB area, the distance between repeatedly superimposed 3D surface models was lower than 1 mm at all seven measurement areas, apart from one measurement at Gonial L, and remained within 0.5 mm in almost all cases. The variability in error measurements was larger at pogonion and gonion, showing that the reproducibility decreased as the distance from the superimposition area increased. Within all methods, no significant differences were detected between the magnitude of error at the various measurement areas (EMACB, Friedman test: *p* = 0.262; TACB, Friedman test: *p* = 0.177; MACB, Friedman test: *p* = 0.805).

The good reproducibility of the EMACB area was confirmed from the Bland–Altman plots that showed adequate agreement between measurements performed at two different time points from the same operator (Figure 3). There was no evidence when using the EMACB method that the differences increased according to the size of the detected T0–T1 changes. Bland–Altman plots testing the reproducibility of the TACB and the MACB area have been previously published [21,22,23].

The agreement between the first and second EMACB superimpositions on entire facial surfaces are visualized in Figure 4, displaying color-coded maps for the average, minimum, and maximum intraoperator errors. The color-coded maps are in accordance with the results provided in the box-plots in Figure 2.

### 3.3. Agreement between EMACB and MACB with TACB

The agreement between the EMACB and MACB methods with TACB method is presented in Figure 5. Overall, there was perfect agreement between the TACB and the EMACB method (median: −0.03 mm, IQR: 0.28 mm, range: −1.51 to 0.72 mm), whereas the agreement between TACB and MACB method was reduced with differences considered in many cases clinically significant (median: −0.15 mm, IQR: 0.83 mm, range: −3.67 to 1.27 mm) (Mann–Whitney test: *p* = 0.004). No systematic error was detected between the TACB and EMACB methods (one sample *t*-test: *p* > 0.05). Systematic error was evident between the TACB and MACB methods (*p* < 0.001). There were no differences between measurement areas when comparing the agreement of the TACB with EMACB method (Friedman test: *p* = 0.074). On the contrary, there were dereferences in the agreement of the TACB with MACB area (Friedman test: *p* < 0.001), considering Gonial R vs. Zygoma R, Gonial R vs. A point, Gonial R vs. N point, and Pogonion vs. N point (Dunn’s test: *p* < 0.05, including Bonferroni adjustment).

The Bland–Altman plots revealed that there was no evidence that the differences between methods increased as the measured T0–T1 distances increased (Figure 6).

Results concerning TACB and EMACB method are confirmed from the color-coded distance maps on Figure 7, showing individual cases representative of the minimum, the average, and the maximum differences of the relocated T1 surface models, superimposed on stable T0 models with both methods. It is evident that the distances did not exceed 1 mm, even in the cases with the maximum differences. The differences between methods can be considered clinically irrelevant.

## 4. Discussion

The present study proposed and tested a novel reference area for the voxel-based superimposition of CBCT images. This work was based on previous investigations evaluating the reliability, accuracy, and reproducibility of the voxel-based superimposition methods in growing individuals [21,22,23,24]. These baseline reports have confirmed the reliability of such methods and have suggested that they can be applied in everyday clinical practice to assess treatment outcomes or morphological changes in three dimensions. However, these studies used relatively large reference areas to perform superimpositions of serial CBCTs. These extended laterally to the middle anterior cranial base structures that remain stable after an early age, onto areas that are changing morphologically during development. The inclusion of the latter areas can skew the superimposition outcomes since the best-fit approximation of morphologically unstable areas will affect the expected perfect overlap of morphologically identical areas in an attempt to minimize the overall differences between corresponding areas/voxels [18].

In order to improve the anatomical accuracy of the selected area for the CBCT superimpositions, a recent study proposed the use of a narrower reference area, which only includes the midlines structures of the anterior cranial base, and thus avoids the inclusion of unstable hard and soft tissues around it [25]. Although the study showed promising results, it did not evaluate the reproducibility of the method in individual patients, because it omitted to test individual differences between repeated measurements. The potential shortcomings of using this narrow reference area were confirmed by a recent investigation that revealed a significant lack of consistency in the superimposition outcomes [23]. The encountered problems may be related to the small reference area, since the rationale of the area selection has a solid biological basis [10]. In the case of 3D surface superimpositions, the selection of small reference areas increases the possibility for artefacts to confound the superimposition outcome [18]. Voxel-based registrations of CBCTs are also based on best-fit registration algorithms, and thus may also be affected similarly by the size of the reference area.

Here, we built upon the rationale of previous research and evaluated the use of a reference area consisting of the midline structures of the anterior cranial base and slightly extended laterally to include the entire width of the anterior clinoid processes (EMACB). Anatomically, this area encompasses the structures that have been well-documented to remain stable after 7 years of age, namely the presphenoid region (sphenoid bone structures anterior to the sella turcica) and the cribriform plate [3,4], and includes less areas that are likely to change over time [10]. Therefore, it was speculated that this novel area might have better anatomical justification as a reference area for the voxel-based superimpositions, while having adequate size to offer robustness to artifacts. The results of the present study reveal that the use of the EMACB area leads to highly reproducible outcomes. The error between repeated superimpositions never exceeded 1 mm and was regularly within the range of 0.5 mm. When testing the reference area initially proposed by Ghoneima et al. [25] and referred to as middle anterior cranial base (MACB), it became evident that it was not reproducible in approximately one third of cases, with differences between repeated superimpositions approaching 2 mm in some instances [23]. To that extent, and in contrast to the EMACB area, the MACB area is not applicable for clinical use due to the reduced consistency considering single patients. It is assumed that the lateral extension to include the entire anterior clinoid processes of the sphenoid bone in the EMACB area provides a more robust reference area. Indeed, a closer look at the direction of the generated error when using the MACB area reveals that this was primarily present on the coronal plane, which implies that the lateral dimension of the reference area was not large enough to provide satisfactory outcomes [23].

When the novel EMACB area was compared to the TACB, which was considered as the reference method, the two methods showed perfect agreement. Individual differences did not exceed 1 mm in any case, which is considered clinically acceptable. Taking into account the small disagreement between the EMACB and TACB and the precision levels of each method detected through reproducibility tests, it can be inferred that the identified disagreements are primarily related to method imprecision. The inclusion of anatomical structures that could have been subjected to changes during the tested period in the TACB method (soft tissues or lateral skeletal structures) might have also contributed to the detected differences to a lesser degree. In any case, the disagreements were small and both methods showed comparable overlap of the stable anterior midline cranial base structures after superimposition, indicating adequate reliability. Taking into consideration that, from an anatomical viewpoint, the EMACB reference area corresponds better to the stable midline structures of the anterior cranial base and avoids areas that may change over time, the present study suggests the EMACB to be used as the reference area of choice when performing the voxel-based superimposition of CBCT volumes.

The sample tested in the present study consisted of 15 pairs of CBCT scans. Sample size is always an important consideration which might affect the robustness of study outcomes in clinical as well as in experimental studies. The present study is a diagnostic accuracy study of an automated method (best-fit algorithm application), where the outcomes of each superimposition for the same pair of CBCTs were compared to each other and individual outcomes were assessed. Thus, if the methods work equally well the expected difference between compared outcomes (e.g., in reproducibility or agreement between methods) should be zero in each single case. In this type of study, a reasonably small number of cases is adequate to prove that the automated diagnostic method performs well. Therefore, a sample size of 15 CBCT pairs fulfils this purpose successfully. This is also supported from empirical evidence in the current literature [9,11,14,16,18,19,21,22,23]. Post hoc power analyses (G*power, version 3.1.9.6) revealed adequate power for comparative statistics (EMACB vs. TACB agreement, power: 97%, a = 0.05; EMACB reproducibility, power: 99%, a = 0.05) to detect an overall difference of 0.5 mm between EMACB and TACB outcomes or between repeated EMACB measurements, which were the primary study outcomes. Regarding the individual areas, the power for both outcomes ranged between 80% and 99% to detect a difference of 0.5 mm at a = 0.05, which is also considered satisfactory.

To eliminate the effect of segmentation error on the superimposition outcomes, the segmentation threshold was kept stable throughout the study by using the automated function of Dolphin Software for extracting skeletal surface models. This increased the precision of the comparisons between repeated superimpositions with the same or different techniques supporting the purpose of the present study, since the threshold level used to perform surface segmentations of 3D radiographic volumes significantly influences the extracted surfaces and may thus confound the superimposition outcomes [14]. The specific segmentation error has been tested previously and is generally considered small [21]; however, it could affect the result of this study if this factor was not controlled.

Although for EMACB and TACB the intraoperator and between-method-error were minimal, it was observed that the error did increase as the distance of measurement area to the reference area increased. Despite the best-fit approximation of serial volumes at the superimposition reference area, small rotations of the entire volume around the reference area are expected, thus leading to a larger error as the distance from the center of rotation increases [28]. This finding was observed in all superimposition methods tested so far and is in agreement to the results of previous investigations [21,22,23].

It must also be noted that all the methods tested here are applicable in large field of view CBCT scans, which include the anterior cranial base. This requires larger radiation doses as well as the exposure of sensitive anatomical structures, such as the pituitary gland and the eyes, to radiation [11,29]. Therefore, despite its validity, the proposed EMACB method is not suggested at present for routine use in clinical settings, unless a clear indication for a large field of view CBCT examination is present. Currently, there are available machines in the market that can produce a large field of view CBCT examination with radiation exposure comparable to the standard radiographic images used for craniofacial diagnosis, but the validity of the present techniques for these low-dose images needs to be investigated. Alternatively, reference areas that are included within limited-field CBCT scans, such as the zygomatic arches, could be used in the future for the superimposition of serial CBCT volumes. Such areas have been studied in the past, but they remain to be adequately validated prior to application [9,11,30].

### 4.1. Limitations

The present sample included CBCT volumes of growing patients, older than eleven years. Since the midline skeletal structures of the anterior cranial base remain stable after the age of seven, the superimposition error reported here cannot be attributed to changes in these structures. Furthermore, due to the stability of the midline anterior cranial base, the results of the study are expected to be applicable also in nongrowing individuals, as well as in individuals older than seven years of age. However, the results are based on CBCT volumes acquired with a particular machine under specific settings. Although these images represent regular quality CBCT images for the assessment of craniofacial morphology, they cannot be generalized to all possible setting configurations and CBCT machines. The effect of image quality on the quality of the superimposition remains to be tested.

### 4.2. Future Research

Future research is needed to confirm and generalize the present findings by applying these methods on CBCT scans acquired from different machines and with different settings. Moreover, the validity of the outcomes could be further confirmed by in vitro studies on dry skulls, where a gold standard method could be applied for comparisons (e.g., high radiation/quality CT scans or direct surface scans of dry skulls where changes have been simulated and serial CBCT scans of the before and after condition were obtained).

## 5. Conclusions

The novel EMACB voxel-based superimposition method described here showed adequate reproducibility in the present growing patient sample and good overall agreement with the previously tested TACB method. Furthermore, the visual assessment of the overlap of the superimposed volumes on the well-established, morphologically stable midline anterior cranial-based structures, in the original radiographic slices, confirmed the validity of the outcomes. Based on the above considerations and the technical, anatomical, and biological principles applied when serial 3D data are superimposed to assess craniofacial changes (use of a stable reference area of adequate size and central location), we recommend the EMACB method as the method of choice to fulfil this purpose.

## Figures and Tables

**Figure 1 jcm-11-03536-f001:**
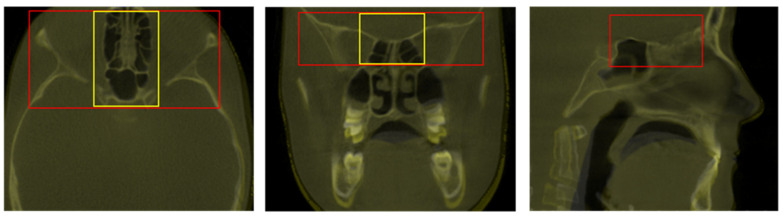
Definition of the Total Anterior Cranial Base area (TACB) (in red) and the Extended Middle Anterior Cranial Base area (EMACB) (in yellow) depicted on T0–T1 volumes, following TACB superimposition. On the right image, the EMACB frame is not visible because it is identical to the one of TACB.

**Figure 2 jcm-11-03536-f002:**
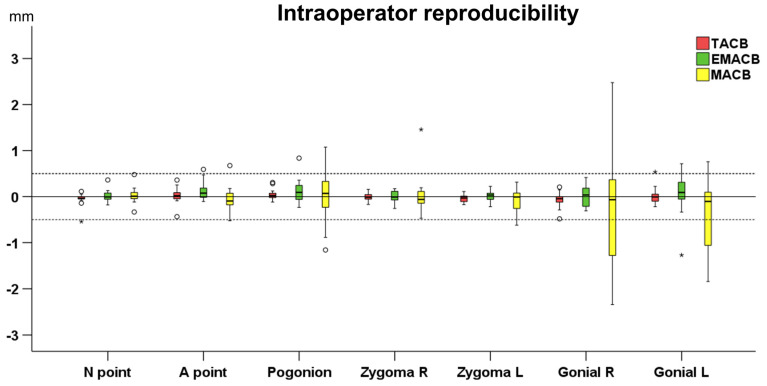
Box plots showing the intraoperator reproducibility of the TACB superimposition (Total Anterior Cranial Base), the MACB (Middle Anterior Cranial Base), and the EMACB (Extended Middle Anterior Cranial Base) on the detected T0–T1 changes in mm, for all measurement areas. Zero value, depicted by the continuous horizontal line, indicates perfect reproducibility, whereas any deviation from zero is considered error. The dashed lines indicate 0.5 mm and −0.5 mm. The upper limit of the black line represents the maximum value, the lower limit the minimum value, the box the interquartile range, and the horizontal black line the median value. Outliers are shown as black circles or stars in more extreme cases.

**Figure 3 jcm-11-03536-f003:**
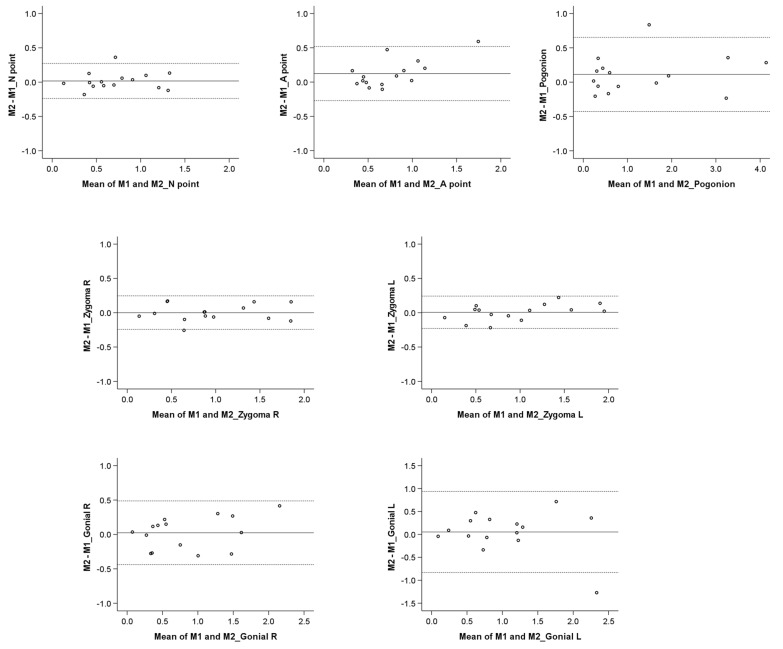
Bland–Altman plots on the T0–T1 changes (mm) detected through repeated EMACB superimpositions by the same operator. The continuous horizontal line shows the mean of the differences in the detected T0–T1 changes, and the dashed lines show the corresponding 95% Limits of Agreement. M1: Measurement 1; M2: Measurement 2.

**Figure 4 jcm-11-03536-f004:**
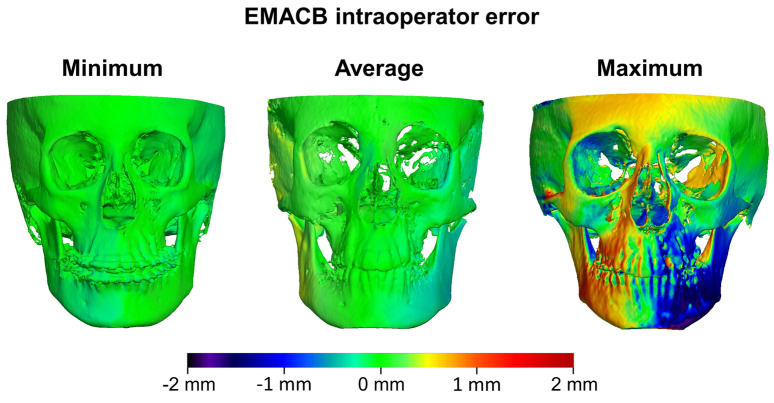
Color coded distance maps showing the intraoperator differences on T1 surfaces obtained from repeated T0–T1 EMACB voxel-based superimpositions, with the T0 surface held constant as a reference. The samples that presented the least (**left**), average (**middle**), and largest (**right**) absolute differences on the seven measurement areas are shown.

**Figure 5 jcm-11-03536-f005:**
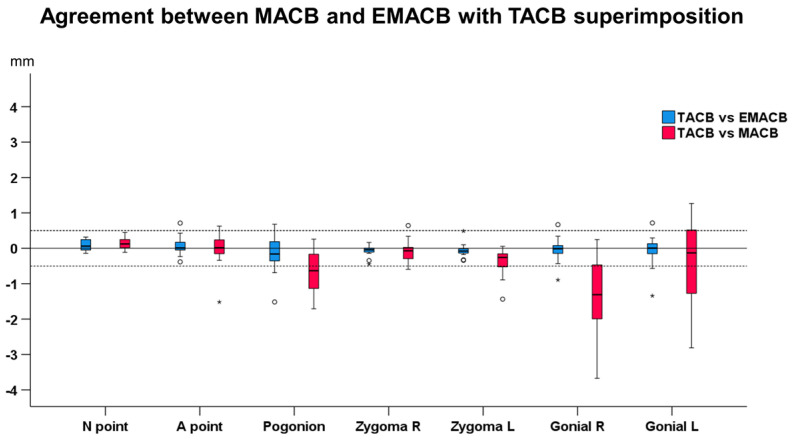
Box plots showing the agreement between EMACB (Extended Middle Anterior Cranial Base) and MACB (Middle Anterior Cranial Base) with the TACB (Total Anterior Cranial Base) superimposition on the detected T0–T1 changes, for all measurement areas. Zero value, depicted by the continuous horizontal line, indicates perfect agreement, whereas any deviation from zero is considered disagreement. The dashed lines indicate 0.5 mm and −0.5 mm. The upper limit of the black line represents the maximum value, the lower limit the minimum value, the box the interquartile range, and the horizontal black line the median value. Outliers are shown as black circles or stars in more extreme cases.

**Figure 6 jcm-11-03536-f006:**
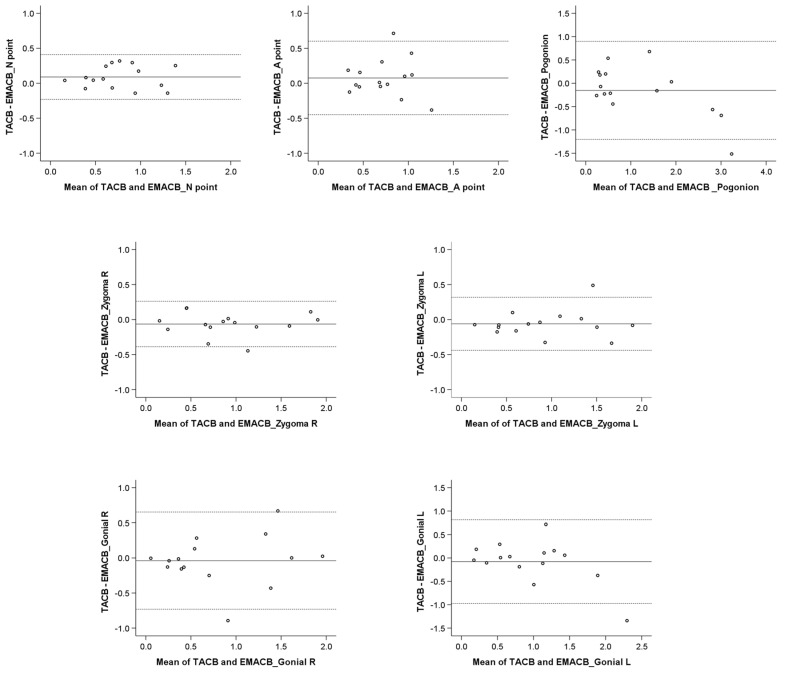
Bland–Altman plots on the T0–T1 changes (mm) detected through the TACB and the EMACB superimpositions by the same operator. The continuous horizontal line shows the mean of the differences in the detected T0–T1 changes, and the dashed lines show the corresponding 95% Limits of Agreement. TACB: Total Anterior Cranial Base; EMACB: Extended Middle Anterior Cranial Base.

**Figure 7 jcm-11-03536-f007:**
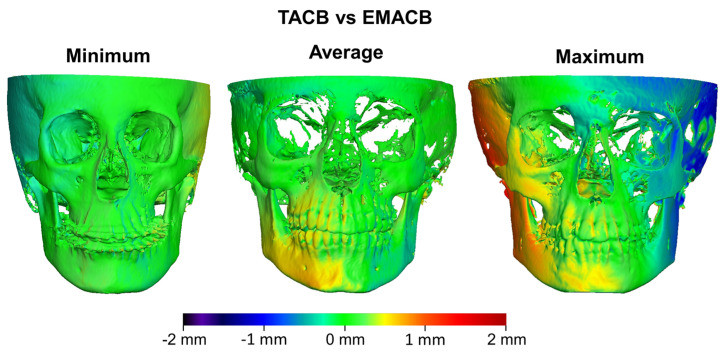
Color coded distance maps showing the differences of T1 surfaces obtained from T0–T1 TACB and EMACB voxel-based superimpositions, with the T0 surface held constant as a reference. The samples that presented the least (**left**), average (**middle**), and largest (**right**) absolute distances of the EMACB from the TACB T1 surface, on the seven measurement areas, are shown.

## Data Availability

All data are available in the main text or the extended data. The protocols and datasets generated and/or analyzed during the current study are available from the corresponding author on reasonable request.

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
