# Peer review of "Novel Anterior Cranial Base Area for Voxel-Based Superimposition of Craniofacial CBCTs"

_jcm, 2022, doi:10.3390/jcm11123536_

Round 1
Reviewer 1 Report
Abstract:
1. Please explain the importance of this research.
2. Please describe the scientific method of comparison.
Introduction
1. CBCT has been widely used in the field of dentistry. Is there any related literature that duplicates the manuscript?
2. The last paragraph proposes to be more specific about the focus of this study.
Materials and Methods
1. 2.3. The sample may be presented in tables or pictures.
2. It is recommended to add the time and parameters of T0 and T1.
3. Please rearrange the paragraph 2.9. Statistical analysis.
Discussion
1. What is the biggest factor that causes the difference between EMACB and TACB?
2. Most of the time I only see you explain where there is a difference or just the result. It is recommended to explain more about the reasons for the difference.
Conclusions
1. Conclusions are similar to the Abstract and suggest modifying the content.
Author Response
Reviewer 1
Reviewer’s comment:
Abstract:
- Please explain the importance of this research.
- Please describe the scientific method of comparison.
Authors’ response: The abstract has been revised as suggested. According to the journal guidelines more technical details should not be provided in the abstract and space considerations do not allow for more extended descriptions.
Reviewer’s comment:
Introduction
- CBCT has been widely used in the field of dentistry. Is there any related literature that duplicates the manuscript?
- The last paragraph proposes to be more specific about the focus of this study.
Authors’ response: 1. To our knowledge, all relevant studies have been cited and discussed in the manuscript. There is no other study testing the suggested reference area (EMACB) until now.
- The following sentence has been added at the end of the introduction section: “We hypothesized that the novel EMACB area is valid, shows adequate agreement to the TACB method, and adequate reproducibility when applied on serial CBCT images of growing patients.”
Reviewer’s comment:
Materials and Methods
- The sample may be presented in tables or pictures.
- It is recommended to add the time and parameters of T0 and T1.
- Please rearrange the paragraph 2.9. Statistical analysis.
Authors’ response: 1. We prefer to keep this information in the text, since the provided data are not extensive.
- This information is provided in sections 2.3 and 2.4 in the submitted manuscript.
- We carefully read section 2.9 and think that the way it is written is acceptable and avoids unnecessary repetitions. If the reviewer insists on changes we would be grateful if we could receive more specific instructions.
Reviewer’s comment:
Discussion
- What is the biggest factor that causes the difference between EMACB and TACB?
- Most of the time I only see you explain where there is a difference or just the result. It is recommended to explain more about the reasons for the difference.
Authors’ response: 1. The following text has been added in the Discussion section: “Taking into account the small disagreement between EMACB and TACB and the precision levels of each method detected through reproducibility tests, it can be inferred that the identified disagreements are primarily related to method imprecision. The inclusion of anatomical structures that could have been subjected to changes during the tested period in the TACB method (soft-tissues or lateral skeletal structures) might have also contributed to the detected differences to a lesser degree. In any case, the disagreements were small and …”
- Thank you very much for this remark. We read the Discussion carefully and performed relevant changes where needed.
Reviewer’s comment:
Conclusions
- Conclusions are similar to the Abstract and suggest modifying the content.
Authors’ response: The conclusion section was adjusted for clarity. We do not consider similarities with the abstract problematic, since they allow increased consistency in reporting.
Reviewer 2 Report
I would like to complement the authors on their hard work in compiling this manuscript. I do have some suggestions to make:
1. Grammatical corrections throughout.
2. Lack of generalizability of findings.
3. Lack of inclusion of adult subjects- possible future recommendations could include adult/non-growing subjects
4. Please add a section on future recommendations
5. I have some concerns regarding the sample size- A suggestion could be to carry out a retrospective audit so that there can be a bigger sample size instead of increasing the samples for a prospective study.
6. Please discuss the power of the sample with its calculation and how can the findings be generalized.
Author Response
Reviewer 2
Reviewer’s comment:
I would like to complement the authors on their hard work in compiling this manuscript. I do have some suggestions to make:
- Grammatical corrections throughout.
Authors’ response: We read the manuscript carefully and corrected any errors detected.
Reviewer’s comment:
- Lack of generalizability of findings.
Authors’ response: This issue has been discussed in section “4.1 Limitations” of the study.
Reviewer’s comment:
- Lack of inclusion of adult subjects- possible future recommendations could include adult/non-growing subjects
Authors’ response: The midline anterior cranial base structures remain stable after approximately 7 years of age. The youngest subject included in the present study was 11 years old at T0. Therefore, when we refer to the midline anterior cranial base in our sample, it is assumed that our statements also apply to adult subjects. For this reason the results of the study are expected to be applicable in non-growing individuals as well, including adults. This information is now provided in the manuscript.
Reviewer’s comment:
- Please add a section on future recommendations
Authors’ response: Following the reviewer’s suggestion, a relevant section “4.2. Future research” has been added at the end of the Discussion section.
Reviewer’s comment:
- I have some concerns regarding the sample size- A suggestion could be to carry out a retrospective audit so that there can be a bigger sample size instead of increasing the samples for a prospective study.
- Please discuss the power of the sample with its calculation and how can the findings be generalized.
Authors’ response: The sample tested in the present study consisted of 15 pairs of CBCT scans. Sample size is always an important consideration which might affect the robustness of study outcomes in clinical as well as in experimental studies. The present study is a diagnostic accuracy study of an automated method (best-fit algorithm application), where the out-comes of each superimposition for the same pair of CBCTs were compared to each other and individual outcomes were assessed. Thus, if the methods work equally well the expected difference between compared outcomes (e.g. in reproducibility or agreement be-tween methods) should be zero, in each single case. In this type of studies, a reasonably small number of cases is adequate to prove that the automated diagnostic method per-forms well. Therefore a sample size of 15 CBCT pairs fulfils this purpose successfully. This is also supported from empirical evidence in the current literature [9,11,14,16,18,19,21-23]. This information is now provided in a paragraph added in the discussion section.
Round 2
Reviewer 1 Report
The manuscript has been moderately revised.
Reviewer 2 Report
There is still a lack of information regarding the power of the sample. Also the references cited as a response for the sample size have a bigger sample size as has been employed by the authors for the purpose of this study.
eg.
1. https://pubmed.ncbi.nlm.nih.gov/32355261/
2. https://onlinelibrary.wiley.com/doi/full/10.1111/j.1469-7580.2011.01346.x
These have been cited by the authors in their response as a measure of the sample size.
